# IL-21 in Conjunction with Anti-CD40 and IL-4 Constitutes a Potent Polyclonal B Cell Stimulator for Monitoring Antigen-Specific Memory B Cells

**DOI:** 10.3390/cells9020433

**Published:** 2020-02-13

**Authors:** Fridolin Franke, Greg A. Kirchenbaum, Stefanie Kuerten, Paul V. Lehmann

**Affiliations:** 1Research & Development Department, Cellular Technology Limited, Shaker Heights, OH 44122, USA; fridolin.franke@fau.de (F.F.); greg.kirchenbaum@immunospot.com (G.A.K.); 2Institute of Anatomy and Cell Biology, Friedrich-Alexander University Erlangen-Nürnberg, 91054 Erlangen, Germany; stefanie.kuerten@fau.de

**Keywords:** antibody-secreting cells, B cell activation, plasmablasts, IgE, ELISPOT, ImmunoSpot, Fluorospot, immunoglobulin classes, influenza, type 1 allergy

## Abstract

Detection of antigen-specific memory B cells for immune monitoring requires their activation, and is commonly accomplished through stimulation with the TLR7/8 agonist R848 and IL-2. To this end, we evaluated whether addition of IL-21 would further enhance this TLR-driven stimulation approach; which it did not. More importantly, as most antigen-specific B cell responses are T cell-driven, we sought to devise a polyclonal B cell stimulation protocol that closely mimics T cell help. Herein, we report that the combination of agonistic anti-CD40, IL-4 and IL-21 affords polyclonal B cell stimulation that was comparable to R848 and IL-2 for detection of influenza-specific memory B cells. An additional advantage of anti-CD40, IL-4 and IL-21 stimulation is the selective activation of IgM^+^ memory B cells, as well as the elicitation of IgE^+^ ASC, which the former fails to do. Thereby, we introduce a protocol that mimics physiological B cell activation through helper T cells, including induction of all Ig classes, for immune monitoring of antigen-specific B cell memory.

## 1. Introduction

The adaptive immune system, on one hand, enables the host to develop long-term immunity to antigens; on the other hand, it allows for fine-tuning of effector functions that are required for successful defense against different challenges. In the case of humoral immune defense, B cells first produce IgM antibodies that are, in large, restricted to the protection of the intervascular compartment. IgM excels in precipitating and neutralizing antigens, as well as in promoting their elimination by phagocytosis. Subsequently, B cells switch to producing IgG and IgA antibodies that can penetrate most tissues of the body and are exported to mucosal surfaces, which extends immune surveillance to the entire body. Under special circumstances, such as persistent antigen exposure, IgE antibodies are also elicited and bind to mast cells and basophils, thereby sensitizing them for local histamine release at the site of antigen encounter. In this chain of events, individual B cells receive specific cytokine cues that promote switching of the antibody type they produce (reviewed in [1]). Importantly, the B cell maintains an identical hypervariable region, which defines antibody specificity, but splices it with a downstream constant region defining a different class/subclass of immunoglobulin. This process of gene rearrangement, called immunoglobulin class switching, is largely under the control of antigen-specific T helper (T_H_) cells, and predominantly occurs within secondary lymphoid structures, such as the spleen and lymph nodes (reviewed in [2]).

Among the T_H_ cells that regulate B cell functions, the follicular helper (T_FH_) subset is critical [3]. T_FH_ cells mediate their effect on B cells via direct cell contact, primarily involving CD40 ligand (CD40L) interactions with CD40 expressed on the B cell, and through targeted cytokine release [4]. The CD40/CD40L interaction induces activation, blastogenesis and proliferation in the B cell without promoting antibody production [5], while various cytokines have been implicated in the terminal differentiation into antibody-secreting plasma cells. T_FH_ are characterized by surface marker expression of CXCR5, PD-1, ICOS, and CD40L, and the production of various cytokines, including IL-4, IL-10, and IL-21 [1]. They localize to germinal centers within the B cell follicle, where they can instruct the maturation of B cells into memory cells, short-lived antibody-secreting plasmablasts or long-lived plasma cells [6,7,8].

Many cytokines have been implicated in the cognate interactions between B and T_H_ cells. These include IL-2 [5,9,10,11], IL-4 [5,12,13,14], IL-6 [9,15,16], IL-10 [5,9,17,18,19], IL-12 [20], IL-13 [21], IL-15 [22], and TGF-β [9,23] and IL-21 [24,25]. IL-2, for example, is known to enhance proliferation and Ig-secretion by activated human B cells [26,27], while IL-4 induces isotype switching preferentially to IgG1, IgG4 and IgE [5,28,29,30,31]; TGF-β promotes class switching to IgA [23]. Among the B cell stimulating cytokines, IL-21 has emerged as a critical regulator of B cell differentiation [6,7,8]. Its dominant expression by T_FH_ cells, along with the upregulation of IL-21R on germinal center B cells, are among many lines of evidence supporting the central role of T_FH_ operating via this cytokine [1]. Moreover, IL-21R is detectable at both the transcript and protein level in B cells and its expression is enhanced following in vitro stimulation through CD40 [32,33]. Purified B cells stimulated with anti-CD40 and anti-IgM also exhibited IL-21 dose-dependent increases in IgG production, and neutralization with an IL-21-Fc fusion protein reduced B cell proliferation and differentiation into antibody-secreting cells (ASC) [34]. Furthermore, upregulation of IL-21R on B cells and expansion of circulating T_FH_-like cells in peripheral blood is a strong predictor of antibody responses following influenza vaccination [35,36]. As such, since IL-21 is centrally involved in B cell differentiation in vivo, we were interested in determining whether it could also be utilized for immune monitoring purposes.

Since polyclonal stimulation is essential for promoting antibody production by class-switched memory B cells in vitro, considerable effort was undertaken to identify suitable substances and protocols, primarily relying on microbial, T cell independent activation stimuli. Pinna et al. [37] compared 60 conditions, including several Toll-like-receptor (TLR) agonists and identified the combination of TLR7/8 agonist R848 and IL-2 as the most potent regimen for in vitro activation of resting human B cells into ASC. They reported that additional stimuli, including CD40L, IL-4, and CpG, promoted only naïve, but not memory B cell activation. This notion was confirmed by Jahnmatz et al. [38], who also systematically compared compounds with known in vitro B cell stimulatory capability, identifying the combination of R848 and IL-2 as the most efficient inducer of ASC. This simple protocol has since evolved as the gold standard for the selective activation of memory B cells for immune monitoring purposes. While this protocol excels in inducing IgM^+^, IgG^+^, and IgA^+^ ASC, it does not elicit IgE^+^ ASC (see below) and more importantly, relying on TLR-induced differentiation does not mimic T cell-dependent activation through T_H_. Of note, neither Pinna’s nor Jahnmatz’ study [37,38] evaluated IL-21 enhancement of in vitro B cell activation.

A protocol that induces IgE^+^ ASC has been identified and requires in vitro stimulation with agonistic anti-CD40 antibody and IL-4 [39]. However, this protocol does not efficiently elicit the differentiation of IgM^+^, IgG^+^ or IgA^+^ ASC (detailed in Results) and therefore is not well-suited for exploration of antigen-specific B cell memory. As IL-21 is a potent B cell growth and differentiation factor, we decided to study here whether its inclusion with R848 and IL-2, or anti-CD40 and IL-4, would constitute a superior polyclonal B cell stimulation system for the monitoring of B cell immunity.

In the context of humoral immunity, antibodies, present in the body, serve as the first line of defense against re-infection. However, antibody molecules are short-lived and require continuous replenishment by plasma cells to maintain protective titers [40]. Often, re-infection or booster vaccination is required to maintain protective antibody levels. If and when pre-existing antibodies are insufficient to prevent re-infection, memory T and B cells generated during the primary adaptive immune response constitute the next line of defense. These quiescent memory cells are poised to rapidly and efficiently respond to antigen re-encounter owing to their increased clonal sizes, tissue distribution, reduced activation thresholds, and prior commitment into specialized effector lineages. In the context of B cells, affinitymaturated antigen receptors and immunoglobulin class-switched memory B cells enhance the efficacy of the secondary antibody response. Owing to these attributes of cellular T and B lymphocyte memory, within a few days of antigen re-encounter immunity can be re-established and amplified to curtail pathogen replication and mitigate clinical disease. Hence, evaluation of cellular immune memory is required for the understanding of host immunity beyond measurement of serum antibody alone.

Immune monitoring specifically aims to establish both the relative frequency and effector function of antigen-specific memory cells in vivo. To this extent, in vitro activation of resting memory B cells is required to reveal their antibody signature. Presently, R848 and IL-2 are commonly utilized for polyclonal stimulation of human B cells, but this approach may not accurately reflect upon a memory B cell’s capacity to participate in a T cell-driven recall response following antigen re-encounter in vivo. In this context, we sought to identify a stimulation protocol that more closely mimics T cell-driven re-activation of memory B cells during a secondary response in vivo.

ELISPOT and Fluorospot assays are uniquely suited to measure antigen-specific antibody production by individual B cells [41]. The membrane is coated with an antigen of interest upon which polyclonally stimulated B cells are seeded. Of all ASC-induced, immunoglobulin will only be efficiently captured around those B cells secreting the antigen-specific antibody, which yields an immunoglobulin “spot” on the membrane. Counting the number of these spots enables assessment of antigen-specific B cell frequencies amongst all the other, non-antigen reactive B cells present in a test cell population, such as peripheral blood mononuclear cells (PBMC). In this way, the relative clonal sizes, that is the magnitude of the antigen-specific fraction of B cells, within the entire B cell pool can be assessed. In a subsequent step of the assay, one stains the membrane-bound (antigen-specific) immunoglobulin (spots) with immunoglobulin class/subclass-specific detection antibodies, permitting identification of the class/subclass of antibody that each individual B cell produced. Multi-color assays can be performed to simultaneously stain for the different Ig classes/subclasses [42]. As the antibody-class/subclass defines the effector function of the respective antibodies, one can assess the ratio of B cells producing the individual antibody types and gain insight into the quality of the antigen-specific memory B cell pool. In another variant of the ELISPOT/Fluorospot assay, instead of an antigen, the membrane is coated with light chain-specific antibodies, which enables capture of any antibody secreted by B cells, irrespective of their antigen specificity. In this way, the frequency of all B cells producing a certain class/subclass of immunoglobulin can be assessed for the entire B cell pool. The advantage of ELISPOT/Fluorospot assays over ELISAs done on culture supernatants is that, in the former, antibody production by B cells is measured at single-cell resolution (defining both number of secreting cells and the amount of antibody produced by each cell as reflected by the spot size/densities) whereas in ELISAs this high content information is lost and only the net antibody production is measured.

Herein, we describe whether IL-21 modulates the antigen-specific and polyclonal B cell response induced by R848 and IL-2 versus anti-CD40 and IL-4 stimulation.

## 2. Materials and Methods

### 2.1. Human Subjects

Peripheral blood mononuclear cells (PBMC) of all 24 human subjects tested in this study were from healthy adults and were selected from the ePBMC^®^ bank of Cellular Technology Ltd. (CTL, Shaker Heights, OH, USA). These subjects were recruited by Hemacare (Van Nuys, CA, USA) and peripheral blood mononuclear cells were isolated by leukopheresis under Hemacare’s IRB. The PBMC were then cryopreserved and stored in liquid nitrogen until testing. Thawing, washing, and counting of the PBMC was performed according to previously described protocols [43], and cells were seeded into polyclonal B cell stimulation cultures within 2 h of thawing.

### 2.2. Polyclonal B Cell Stimulation

For polyclonal stimulation, the freshly thawed ePBMC^®^ were resuspended in complete B cell medium (BCM) containing RPMI 1640 (Lonza, Walkersville, MD, USA) supplemented with 10% fetal bovine serum (Gemini Bioproducts, West Sacramento, CA, USA), 100 U/mL penicillin/streptomycin, 2 mM L-Glutamine, 1 mM sodium pyruvate, 8 mM HEPES (Life Technologies, Grand Island, NY, USA) and 50 µM beta-mercaptoethanol (Sigma-Aldrich, St. Louis, MO, USA). R848 (Enzo Life Sciences, Ann Arbor, MI, USA) and recombinant human IL-2 (Peprotech Inc., Rocky Hill, NJ, USA) were used at a final concentration of 1 µg/mL and 10 ng/mL, respectively. Anti-CD40 (BioLegend, San Diego, CA, USA), recombinant human IL-4 and IL-21 (R&D Systems, Minneapolis, MN, USA) were used at final concentrations of 1 µg/mL, 30 ng/mL, and 50 ng/mL, respectively. The ePBMC^®^ were cultured at 2 × 10^6^ cells/mL, in either 24-well Greiner CELLSTAR^®^ suspension culture plates or 25 cm^2^ tissue culture flasks (Corning, Sigma-Aldrich), and were incubated at 37 °C, 5% CO_2_ for five days or as otherwise specified in the figure legends. Of note, an increased frequency of B cells (CD19^+^) following stimulation with R848 and IL-2 or anti-CD40, IL-4 and IL-21 relative to media was confirmed by flow cytometry (Appendix A).

### 2.3. B Cell ImmunoSpot^®^ Assays

After polyclonal stimulation, the cells were counted using CTL’s Live/Dead cell counting suite on an ImmunoSpot^®^ S6 Ultimate Analyzer (Cellular Technology Limited, Shaker Heights, OH, USA). After washing with phosphate-buffered saline (PBS), cell pellets were resuspended at 2 × 10^5^ live cells/mL or 3 × 10^6^ live cells/mL in BCM and used immediately in ImmunoSpot^®^ assays. For detecting all ASC, irrespective of their antigen specificity, cell suspensions were serially diluted two-fold in duplicates, starting at 2 × 10^4^ live cells per well, in round-bottom 96-well tissue culture plates (Corning, Sigma-Aldrich) and subsequently transferred into assay plates pre-coated with anti-κ/λ capture antibody contained in the human IgM/IgG/IgA Three-Color ImmunoSpot^®^ kit (from CTL). The cells were cultured for 16 h at 37 °C, 5% CO_2_, and plate-bound Ig spot-forming units (SFU), each representing the secretory foot-print of a single ASC, were visualized using the IgM-, IgG-, and IgA-specific detection reagents contained in the kit, which were used according to the manufacturer’s instructions. For detecting antigen-specific ASC, the Three-Color antigen-specific ImmunoSpot^®^ assays were performed similarly, except that the cells were seeded into antigen-coated assay plates at 3 × 10^5^ live cells/well or 5 × 10^4^ live cells/well. To achieve optimal antigen coating, assay plates were initially precoated with purified anti-His tag antibody (BioLegend) at 10 μg/mL overnight at 4 °C, followed by the addition of recombinant hemagglutinin (rHA) proteins representing the A/California/07/2009 (H1N1) or A/Texas/50/2012 (H3N2) influenza vaccine strains, used at 10 µg/mL for 24 h at 4 °C, followed by one wash with 150 μL PBS, and blocking with BCM for 1h at room temperature. Both rHA influenza antigens were a gift of Dr. Ted Ross (UGA, Athens, GA, USA). For enumeration of IgE^+^ ASC, polyclonally-stimulated PBMC suspensions were serially diluted three-fold in triplicates, starting at 3 × 10^5^ live cells per well, in round-bottom 96-well tissue culture plates and subsequently transferred into anti-κ/λ capture antibody coated plates. The human IgE Single-Color ImmunoSpot^®^ detection kit (from CTL) was then used to visualize IgE^+^ ASC according to the manufacturer’s instructions. Following completion of B cell ImmunoSpot^®^/Fluorospot assay detection systems, plates were air-dried prior to scanning on an ImmunoSpot^®^ S6 Ultimate Reader. The spot-forming units per well (SFU/well) were determined using the Basic Count mode of the ImmunoSpot^®^ Software (Version 7.0.26.0). For each ePBMC^®^ donor and Ig class, SFU counts were determined for each input from the dilution series, and the SFU/well counts for samples were extrapolated to SFU/10^5^ cells per well. As ImmunoSpot^®^ Multi-color B cell kits, analyzers, and software proprietary to CTL were used in this study; hereafter we refer to the ELISPOT/Fluorospot methodology as ImmunoSpot^®^.

### 2.4. ELISA Assays

Costar^®^ 96-well EIA/RIA assay microplates (Sigma-Aldrich) were coated with an optimized concentration of polyclonal capture reagents for human IgM, IgG, IgA or IgE (all from CTL) overnight at 4 °C. The plates were then blocked with ELISA blocking buffer containing 2% *w*/*v* bovine serum albumin in PBS with 0.1% *v*/*v* Tween20 (PBS-T) (Sigma-Aldrich) for 1 h at room temperature. Supernatants of the polyclonally stimulated B cells were then serially diluted and incubated for 2 h, followed by four washes with PBS prior to addition of horseradish peroxidase-conjugated anti-human IgM, IgG, or IgA detection reagents (all from CTL). For quantification of IgE, biotinylated anti-human IgE mAb was used, followed by the addition of HRP-conjugated streptavidin (all from CTL). Secondary detection reagents were incubated for 2 h, after which plates were washed four times with PBS and then developed by addition of TMB chromogen solution (Thermo Fisher Scientific, Waltham, MA, USA). 1M HCl was used to stop conversion of TMB and optical density then measured at 450 nm (OD_450_) using a Spectra Max 190 plate reader (Molecular Devices, San Jose, CA, USA). Concentrations of antibodies were interpolated based on standard curves generated using reference proteins: polyclonal human IgM, IgG, IgA reference proteins, or human IgE mAb were obtained from Athens Research and Technology (Athens, GA, USA) or EMD Millipore (Burlington, MA, USA), respectively.

### 2.5. Statistical Methods

Student’s *t*-test was used to evaluate differences between groups (GraphPad Prism 8, San Diego, CA, USA).

## 3. Results

### 3.1. IL-21 Has Marginal Effects on Polyclonal IgM Production

In the first set of experiments, we tested if including IL-21 into the classic B cell stimulating protocol, relying on the TLR7/8 ligand R848 + IL-2, would enhance the production of IgM antibodies. PBMC from healthy human donors (*n* = 24) were stimulated under both conditions and the induction of IgM measured both in culture supernatants by ELISA (Figure 1A) and at the level of single cells using ImmunoSpot^®^ (Figure 1B). While an increased abundance of IgM was observed with addition of IL-21 to R848 + IL-2 (Condition 3) by ELISA, it did not reach the level of significance (*p* = 0.059) for total IgM^+^ ASC. Therefore, a comparable number of IgM^+^ ASC was induced in the presence or absence of IL-21, however, the per cell productivity was increased in the presence of this cytokine.

In comparison to the classical R848 + IL-2 protocol, neither anti-CD40 + IL-4 (Condition 4), anti-CD40 + IL-4 + IL-21 (Condition 5) or IL-21 alone (Condition 6) induced robust levels of IgM in culture supernatants (Figure 1A). Overall, this outcome was also reflected at the level of IgM^+^ ASC by ImmunoSpot^®^. Thus, each of the T cell relevant conditions (C4, C5, C6) triggered a fraction of IgM^+^ ASC compared to R848 + IL-2 ± IL-21. However, both the ELISA and ImmunoSpot^®^ data also evidenced that anti-CD40 + IL-4 + IL-21 (Condition 5) elicited a significantly increased population of IgM^+^ ASC relative to anti-CD40 + IL-4 (Condition 4) or IL-21 alone (Condition 6) (Figure 1). These IgM^+^ ASC very likely entail antigen-experienced IgM^+^ B cells (as will be detailed below), suggesting that anti-CD40 + IL-4 + IL-21, unlike the R848 + IL-2 protocol, does not trigger terminal differentiation of naïve B cells. Collectively, these data indicate that TLR-driven B cell activation by R848 + IL-2 is superior for the induction of IgM^+^ ASC; however, this outcome may not be pertinent to immune monitoring because the majority of IgM^+^ precursor cells elicited by this stimulation approach will belong to the naïve B cell subset.

### 3.2. IL-21 Triggers IgG and IgA Production in Conjunction with anti-CD40 + IL-4

PBMC stimulated with R848 + IL-2 ± IL-21 yielded robust IgG responses, as measured by both ELISA and ImmunoSpot^®^ (Figure 2A,B). However, there was no further increase in the abundance of IgG or IgG^+^ ASC with addition of IL-21 (Condition 3 vs. 2). Additionally, stimulation with anti-CD40 + IL-4 + IL-21 (Condition 5) also elicited a strong IgG response, as measured by ELISA and ImmunoSpot^®^. Strikingly, stimulation with IL-21 alone (Condition 6) was sufficient to trigger an IgG^+^ ASC response, although it was significantly reduced in magnitude compared to the IgG^+^ response generated following anti-CD40 + IL-4 + IL-21 stimulation (Condition 5) (Figure 2). The IgG^+^ ASC response triggered by IL-21 alone (Condition 6) may represent recently activated memory B cells, whereas additional T cell helper-derived signals provided by anti-CD40 + IL-4 (Condition 5) may be required to promote antibody secretion by quiescent memory B cells. Notably, and most relevant to this communication, the number of IgG^+^ ASC elicited under Conditions 2 or 5 were not statistically different (*p* = 0.071), although the abundance of IgG in supernatants was increased following stimulation under Condition 5 by ELISA.

Similar to the results obtained for IgG, there was no significant enhancement in the IgA^+^ ASC response when IL-21 was included with R848 + IL-2 (Condition 3 vs. 2) by ELISA or ImmunoSpot^®^ (Figure 3A,B). Moreover, the combination of anti-CD40 + IL-4 + IL-21 (Condition 5) also yielded a robust IgA response, whereas anti-CD40 + IL-4 (Condition 4) or IL-21 alone (Condition 6) failed to stimulate a response significantly greater than medium alone (Condition 1) (Figure 3). Furthermore, there was no difference in the abundance of IgA and IgA^+^ ASC produced under R848 + IL-2 (Condition 2) or anti-CD40 + IL-4 + IL-21 (Condition 5) by ELISA and ImmunoSpot^®^. Collectively, these data highlight that both R848 + IL-2 (Condition 2) and anti-CD40 + IL-4 + IL-21 (Condition 5) constitute potent polyclonal stimulators of class-switched memory B cells.

### 3.3. IL-21 Potentiates IgE Production Elicited by Anti-CD40 + IL-4

While IgM^+^, IgG^+^ and IgA^+^ ASC can readily be detected in PBMC following in vitro stimulation with R848 + IL-2, IgE^+^ ASC were not induced under this condition irrespective of whether IL-21 was also included (data not shown). To detect IgE^+^ ASC, a protocol has become prevalent that relies on anti-CD40 + IL-4 stimulation. As shown above, this protocol (Condition 4) has a limited capacity to elicit IgM^+^, IgG^+^ and IgA^+^ ASC. Therefore, and in light of the enhancements observed for the other Ig classes, we tested whether addition of IL-21 would improve the IgE response induced by anti-CD40 + IL-4 stimulation (Condition 5). IgE abundance was significantly increased following stimulation under anti-CD40 + IL-4 + IL-21 (Condition 5) compared to anti-CD40 + IL-4 alone (Condition 4) by ELISA (Figure 4A). Moreover, a significant increase in IgE^+^ ASC was also evidenced at day 5 and day 7 using ImmunoSpot^®^ (Figure 4B,C). As illustrated by the representative images presented in Figure 4d, the addition of IL-21 to anti-CD40 + IL-4 (Condition 5) dramatically enhanced the numbers of IgE^+^ ASC as well as their IgE production rate per B cell compared to anti-CD40 + IL-4 alone (Condition 4). Thus, and relevant to this communication, the anti-CD40 + IL-4 + IL-21 stimulation approach constitutes a major advantage over the traditional R848 + IL-2 protocol through its capacity to elicit all Ig classes, including IgE.

### 3.4. Effect of IL-21 on the Antigen-Specific B Cell Response

The previous data focused on the effect of IL-21 at the level of total immunglobulin class secretion following polyclonal B cell stimulation. Since the major focus of B cell immune monitoring is to evaluate antigen-specific memory, and because memory B cells for individual antigens constitute a minor fraction of the entire B cell pool (that may have differential requirements than naïve B cells for antibody secretion), we next sought to compare the ability of R848 + IL-2 (Condition 2) and anti-CD40 + IL-4 + IL-21 (Condition 5) to recall antigen-specific memory B cells. We selected two influenza antigens, CA/09 (H1N1) and TX/12 (H3N2), as a model for re-activation of antigen-specific memory B cell reactivity for which our test population would have been primed through natural exposure and/or vaccination. Of note, since anti-CD40 + IL-4 alone did not efficiently induce ASC (see Figure 1, Figure 2 and Figure 3), and because no apparent enhancement was observed when IL-21 was added to R848 + IL-2, both of these conditions were not tested.

A side-by-side comparison of PBMC stimulated with R848 + IL-2 (Condition 2) or anti-CD40 + IL-4 + IL-21 (Condition 5) revealed comparable frequencies of class-switched (IgG^+^ or IgA^+^) ASC reactive with the CA/09 and TX/12 rHA antigens (Figure 5B,C,E,F). Of note, controls confirmed the specificity of the antigen-specific Fluorospot assay since negligible spot formation was detected on bovine serum albumin or irrelevant mouse IgG1, κ antigens while spots were evident in wells coated with either of the recombinant influenza antigens (Appendix A). Collectively, this highlights the specificity of the observed responses and supports the notion that they originate from pre-existing class-switched memory B cells.

With regard to IgM^+^ ASC, PBMC stimulated with anti-CD40 + IL-4 + IL-21 (Condition 5) yielded an increased frequency of CA09- or TX/12 rHA-reactive cells relative to PBMC stimulated with R848 + IL-2 (Condition 2). However, it is important to note that stimulation with anti-CD40 + IL-4 + IL-21 yielded only a fraction of the total IgM^+^ ASC response in comparison to R848 + IL-2 (Figure 1). Therefore, the increased frequency of CA/09 or TX/12-reactive IgM^+^ ASC following stimulation with anti-CD40 + IL-4 + IL-21 is consistent with selective activation of IgM^+^ memory B cells. Collectively, these data demonstrate that in the context of immune monitoring, either R848 + IL-2 or anti-CD40 + IL-4 + IL-21 are equally well-suited for in vitro recall of antigen-specific, class-switched (IgG^+^ or IgA^+^) memory B cells; however, anti-CD40 + IL-4 + IL-21 stimulation may enable selective monitoring of antigen-experienced IgM^+^ memory cells without concomitant activation of the naïve B cell subset.

## 4. Discussion

The data presented here suggest that IL-21 has no major effect on IgM, IgG and IgA production when combined with R848 and IL-2. However, IL-21 in conjunction with anti-CD40 and IL-4 constitutes an alternative protocol that is capable of triggering full-blown activation of B cells into ASC. Of note, anti-CD40, IL-4, and IL-21 yielded a comparable IgG^+^ and IgA^+^ ASC response compared to R848 and IL-2, although the latter stimulated significantly increased IgM^+^ ASC. Importantly, comparable influenza-specific IgG^+^ and IgA^+^ ASC responses were detected following in vitro activation of donor PBMC under either condition. However, the frequency of influenza-specific IgM^+^ ASC was significantly increased following anti-CD40, IL-4 and IL-21 treatment. Collectively, our data indicate that anti-CD40, IL-4, and IL-21 is a potent polyclonal stimulation regimen that preferentially stimulates antibody production by memory B cells alone, whereas R848 and IL-2 stimulate antibody production by naïve B cells as well.

Our studies evaluated the usage of IL-21, a T_FH_-derived cytokine, for in vitro stimulation of human B cells in the context of immune monitoring. However, it should be noted that IL-21 is a pleiotropic cytokine with complex activities in vivo [44]. Additionally, IL-21 is secreted by other helper T cell subsets and acts on a variety of innate and adaptive immune cell subsets [45]. Accordingly, IL-21′s activities may serve to either promote or inhibit tumor progression. Paradoxically, induction of antibody responses against p53 has been linked to a poor prognosis in lung cancer [46]. Therefore, T_FH_-dependent activation of antigen-specific B cells, which is essential for host defense against invading organisms such as viruses and bacteria, may have adverse consequences in the context of cancer.

Several studies have reported that IL-21 in conjunction with CD40 signaling triggers differentiation of human or mouse B cells [32,47,48,49,50]. Ding et al. [51] studied the mechanism of CD40 and IL-21 receptor stimulation synergism on the level of intercellular signaling. This work showed that IL-21 alone can induce low-level Blimp-1 expression, a transcriptional regulator of terminal differentiation into ASC; however, maximum Blimp-1 induction required both IL-21R and CD40 ligation. They also showed that CD40L on its own has no effect on Blimp-1 expression, but greatly amplifies the duration of IL-21 induced Jak-STAT3 signaling. Moreover, Dam et al. [33] reported a correlation between IL-21R expression on peripheral B cells and the induction of pSTAT3 following in vitro stimulation with IL-21. Collectively, these data are in line with the observations we report here through measuring downstream antibody production as the readout following five days of stimulation.

The primary objective of immune monitoring efforts is to determine whether there is evidence for prior antigen exposure through the detection of an effector cell response. Secondarily, immune monitoring seeks to define the quality and durability of in vivo primed memory cell populations through measurement of their response profiles following antigen re-encounter. In the context of B cells, this entails measurement of pre-existing antibody that would confer immediate protection upon re-exposure and detection of expanded antigen-specific precursor frequencies. Since memory B cells are preserved in a resting state and do not exhibit constitutive antibody secretion, in vitro stimulation is necessary to promote their conversion into antibody-secreting plasmablasts. Preserving the relative frequencies and immunoglobulin class distribution patterns of these antigen-specific B cells during this process is, therefore, of utmost importance. However, several reports have suggested that acquisition of the CD27^++^ CD38^++^ surface phenotype and differentiation into ASC requires multiple cell divisions [52,53]. Therefore, there exist the possibility that relative frequencies of particular Ig classes (IgM, IgG, IgA, IgE), especially amongst antigen-specific populations, could be skewed during the stimulation interval. In this regard, kinetic tracking studies of antigen-specific B cell reactivity using replicate donor PBMC inputs are required to precisely determine the impact of stimulation duration.

In our protocol, we included IL-4 in addition to anti-CD40 and IL-21. In a systematic study, IL-4 was found to inhibit IgM production triggered by IL-21 in naïve B cells, while CD40 + IL-21 stimulated memory B cells to secrete antibodies irrespective of IL-4′s presence [50]. This notion is essential for immune monitoring since the latter aims at detecting in vivo primed memory B cells without interference by in vitro primed naïve cells. Therefore, the inhibitory effect of IL-4 on in vitro differentiation of naïve B cells is important because CD40 + IL-21 stimulation triggers proliferative responses by naïve B cells to the same extent as memory B cells [32,48,50]. Furthermore, inhibiting the in vitro activation of naïve B cells through the addition of IL-4 is also important in the context of immune monitoring because IL-21 in conjunction with CD40 stimulation induces class-switching [47,48].

One striking observation generated in context of our own work was that IL-21 alone can induce a subpopulation of B cells to convert to IgG^+^ ASC (Figure 2C6), whereas no IgM^+^ (Figure 1C6) and IgA^+^ (Figure 3C6) ASC responses were detected in parallel. This result is particularly interesting from the perspective of immune monitoring. Whether these IL-21 sensitive IgG^+^ B cells overlap with recently activated B cells defined by CD71 expression [54] will require further examination. Moreover, the observation that IgM^+^ and IgA^+^ memory B cells required anti-CD40 stimulation beyond IL-21 alone for terminal differentiation, suggests that an underlying genetic programing difference may exist between IgG vs. IgM and IgA memory B cells.

We relied on two experimental readouts for studying the effect of IL-21 on in vitro differentiation of B cells, namely ELISA and ImmunoSpot^®^. Both techniques provided complementary results when comparing the magnitude of the overall antibody secretion semi-quantitatively. Thus, both techniques verified our major conclusion that IL-21 has an enhancing effect on the differentiation of IgM^+^, IgG^+^, IgA^+^, and IgE^+^ ASC following in vitro stimulation with anti-CD40 and IL-4. However, the ImmunoSpot^®^ assay approach provided additional information that was undetectable by ELISA. Specifically, that a low frequency of IgG^+^ ASC was in fact triggered by IL-21 alone (Figure 2C6). Compared to ELISA, the ImmunoSpot^®^ approach is orders of magnitude more sensitive for detecting analyte secretion by rare cells, in particular when their per cell productivity is low. In ImmunoSpot^®^, the analyte is captured in close proximity to the secreting cell before it can be diluted into the culture supernatant, leaving a footprint for each secreting cell, while in ELISA the analyte is measured in the culture supernatant after it underwent dilution. Therefore, the ImmunoSpot^®^ platform permits the study of rare sub-populations of ASC, which include antigen-specific memory B cells that may otherwise go undetected using alternative detection systems.

Culturing B cells in the presence of anti-CD40, IL-4 and IL-21 was found to trigger influenza antigen-specific ASC in frequencies similar to R848 and IL-2 stimulation (Figure 5). While both polyclonal B cell stimulating protocols apparently are comparable in their utility for immune monitoring purposes, they signify fundamentally different immune biological processes. R848 is a TLR agonist that stimulates B cells via archaic pattern recognition receptors in a T cell-independent manner, whereas stimulation with anti-CD40, IL-4, and IL-21 closely mimics a T cell-dependent instructive program. While neither of these stimulation approaches involve direct B cell receptor engagement, which in vivo is a central component of cognate T cell/B cell interactions, both are capable of eliciting full-blown ASC induction. However, it remains to be determined whether the former (R848 and IL-2) accurately reflects upon a memory B cell’s capacity to participate in a secondary immune response in vivo. In this context, in vitro stimulation of B cells with anti-CD40, IL-4 and IL-21 may more accurately predict this outcome and may therefore hold greater promise for immune monitoring efforts. Future efforts, beyond the scope of this communication, will also delve into the IgG subclass usage of antigen-specific memory B cell populations comparing the outcomes of R848 and IL-2 vs. anti-CD40, IL-4 and IL-21 stimulation to specifically address the relevance of in vitro class-switching by the latter.

Lastly, a striking observation presented in this communication was the elicitation of an IgE^+^ ASC response after anti-CD40, IL-4 and IL-21 stimulation that was undetectable after R848 and IL-2 exposure of the same donor PBMC (data not shown). As mentioned before, CD40 and IL-21R ligation can induce immunoglobulin class-switching in naïve B cells [50]. Additionally, Milovanovic et al. [55] reported that supplementation of IL-17A to anti-CD40 and IL-4 stimulation enhanced the abundance of IgE and IgE^+^ ASC following in vitro stimulation of purified human B cells. Moreover, they also demonstrated that stimulation with anti-CD40 and IL-4, or IL-17A alone, significantly increased mRNA expression levels of both AID and germline transcripts of the IgE (epsilon) constant region. Therefore, one possibility is that the IgE antibodies detected in our studies were produced as a consequence of class-switching by naïve B cells in vitro [56]; in which case these antibodies would not reflect on IgE^+^ B cell memory in vivo. However, even in this scenario, the utilization of the anti-CD40, IL-4 and IL-21 stimulation protocol provides a much more efficient means for inducing IgE^+^ ASC compared to the anti-CD40 and IL-4 protocol that has been used for this purpose previously ([56] and Figure 4). As IL-4 was shown to inhibit IL-21-driven antibody secretion by naïve B cells, it is also possible that the IgE^+^ ASC elicited by anti-CD40, IL-4 and IL-21 arise from antigen-experienced memory B cells that had already undergone class-switching in vivo. In this regard, the recent population by Hoof et al. [57] is particularly relevant, since their studies revealed a shared V_H_ repertoire between IgG^+^ memory cells (termed IgG_E_) and IgE^+^ ASC through single-cell transcriptomics. The overarching conclusion from this work was that antigen-specific (grass pollen) IgE^+^ ASC originated from affinity-matured IgG^+^ cells that were already pre-committed to class-switch to IgE following antigen encounter. Thus, anti-CD40, IL-4 and IL-21 (or IL-17A) stimulation may be a viable strategy for detecting such antigen-specific memory B cell populations that are pre-committed to the IgE ASC fate. Moreover, comparison of IgE^+^ ASC induction using PBMC from atopic and non-atopic donors collected at multiple time-points throughout the year, in which allergen exposure would be more or less likely, would also be informative for determining if pre-committed IgE^+^ ASC precursors are consistently present, or whether this cell population is transiently induced in close proximity to allergen encounter. Furthermore, since detection of IgE^+^ human memory B cells is technically challenging using a flow cytometric approach [58,59], the ability to monitor IgE responses more precisely through deliberate in vitro stimulation would constitute a breakthrough.

## Figures and Tables

**Figure 1 cells-09-00433-f001:**
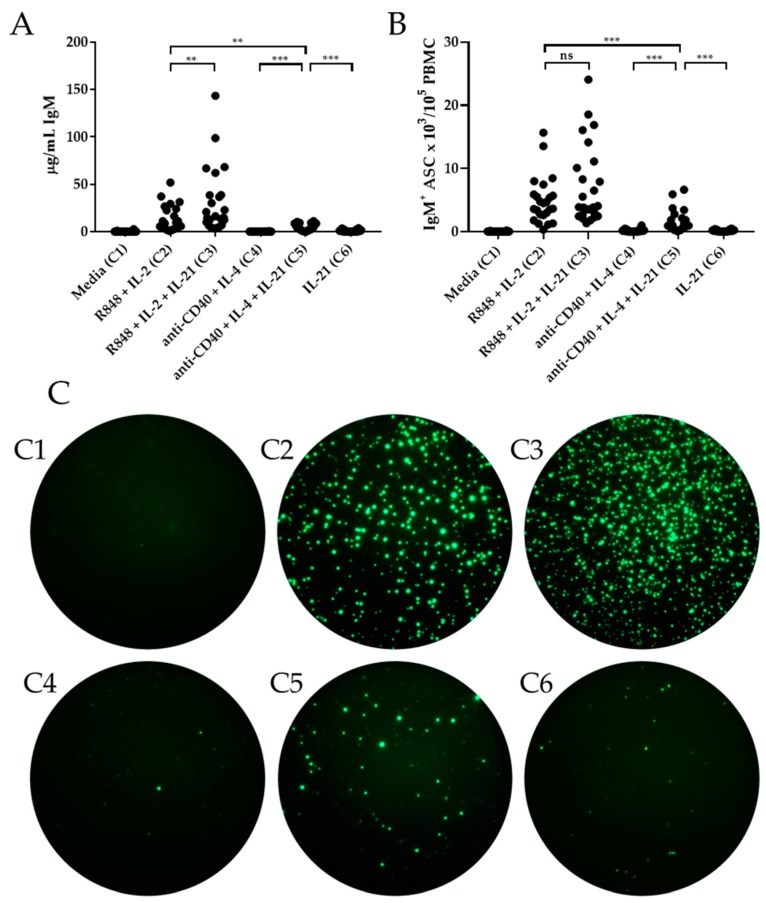
IL-21′s effect on in vitro induction of IgM^+^ ASC. Healthy human donors (*n* = 24) were stimulated in vitro under the specified six culture conditions (detailed in Materials and Methods) and the induction of IgM^+^ ASC evaluated by ELISA (panel A) or ImmunoSpot^®^ (panels B and C). (**A**) Abundance of IgM in culture supernatants following 11 days of in vitro stimulation. (**B**) IgM^+^ ASC were enumerated by ImmunoSpot^®^ following five days of in vitro stimulation. (**C**) Representative ImmunoSpot^®^ well images, containing 10^4^ live cells per well, from a single donor (Donor 386) following in vitro stimulation. C1–C6 correspond to the stimulation conditions specified on the *x*-axis of panels A and B. * *p* < 0.05, ** *p* < 0.01, and *** *p* < 0.001.

**Figure 2 cells-09-00433-f002:**
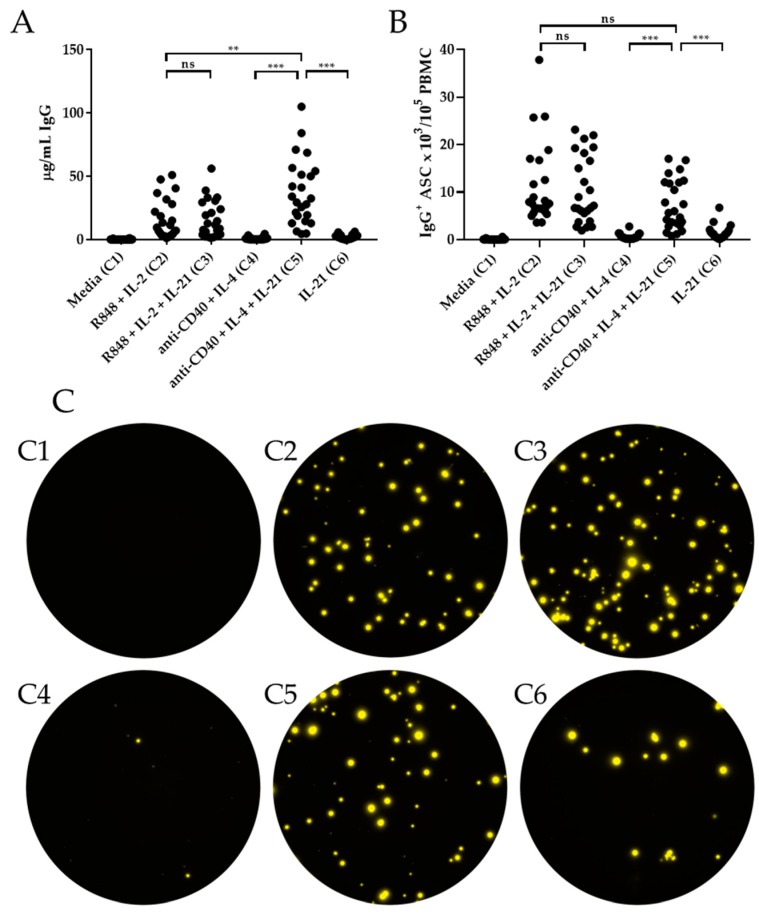
IL-21′s effect on in vitro induction of IgG^+^ ASC. Healthy human donors (*n* = 24) were stimulated in vitro under the specified six culture conditions (detailed in Materials and Methods) and induction of IgG^+^ ASC evaluated by ELISA (panel A) or ImmunoSpot^®^ (panels B and C). (**A**) Abundance of IgG in culture supernatants following 11 days of in vitro stimulation. (**B**) IgG^+^ ASC were enumerated by ImmunoSpot^®^ following five days of in vitro stimulation. (**C**) Representative ImmunoSpot^®^ well images, containing 1,250 live cells per well, from a single donor (Donor 386) following in vitro stimulation. C1–C6 correspond to the stimulation conditions specified on the *x*-axis of panels A and B. * *p* < 0.05, ** *p* < 0.01, and *** *p* < 0.001.

**Figure 3 cells-09-00433-f003:**
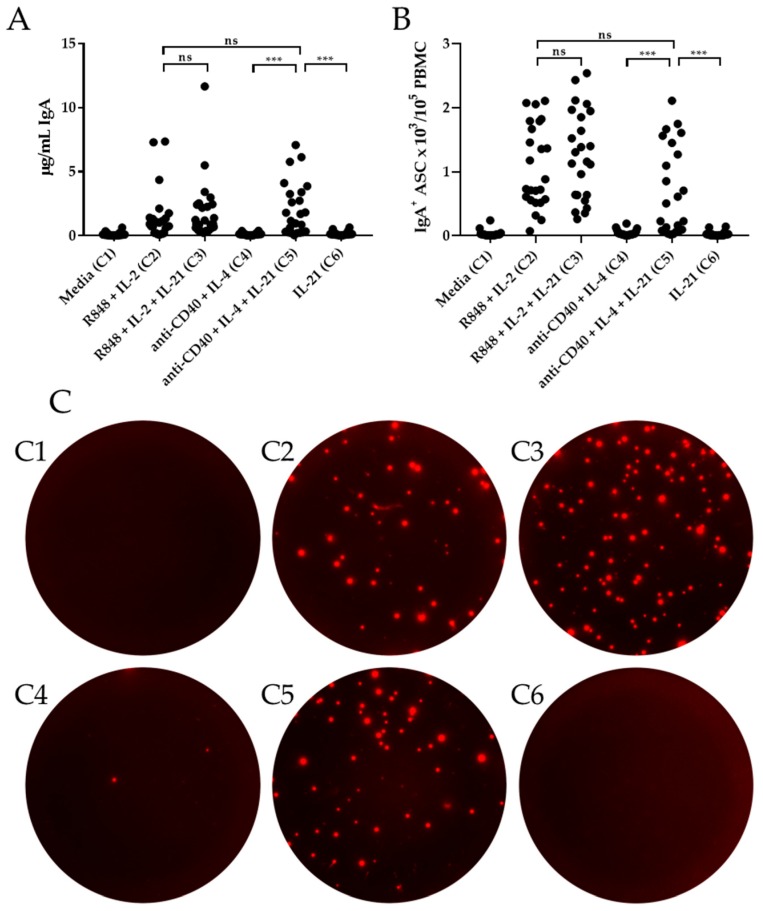
IL-21′s effect on in vitro induction of IgA^+^ ASC. Healthy human donors (*n* = 24) were stimulated in vitro under the specified six culture conditions (detailed in *Materials and Methods)* and induction of IgA^+^ ASC evaluated by ELISA (panel A) or ImmunoSpot^®^ (panels B and C). (**A**) Abundance of IgA in culture supernatants following 11 days of in vitro stimulation was assessed by ELISA. (**B**) IgA^+^ ASC were enumerated by ImmunoSpot^®^ following five days of in vitro. (**C**) Representative ImmunoSpot^®^ well images, containing 10^4^ live cells per well, from a single donor (Donor 386) following in vitro stimulation. C1-C6 correspond to the stimulation conditions specified on the *x*-axis of panels A and B. * *p* < 0.05, ** *p* < 0.01, and *** *p* < 0.001.

**Figure 4 cells-09-00433-f004:**
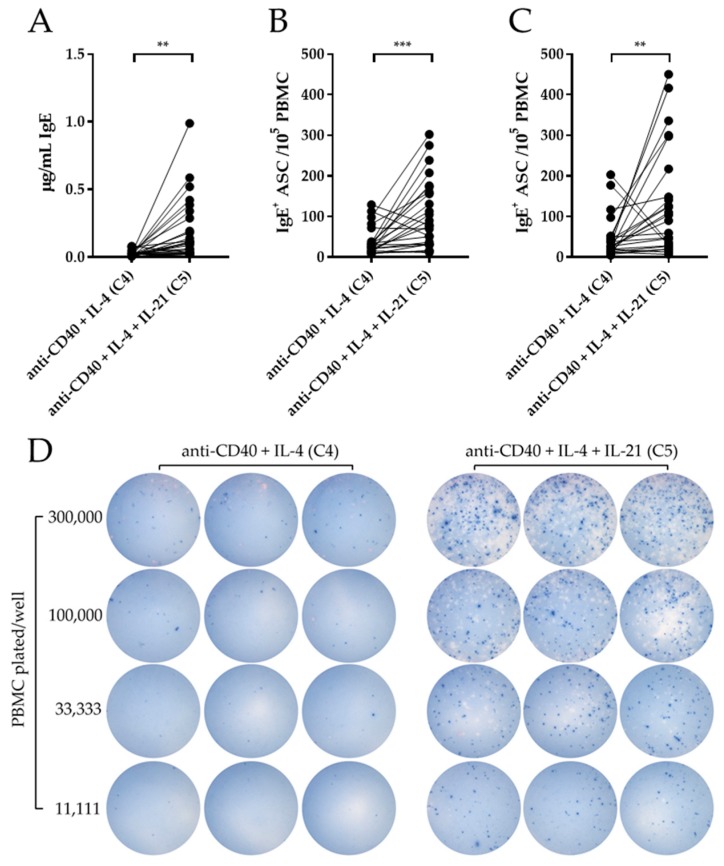
IL-21′s effect on in vitro induction of IgE^+^ ASC. Healthy human donors (*n* = 24) were stimulated in vitro under the specified conditions (detailed in *Materials and Methods*). (**A**) Abundance of IgE in culture supernatants following 11 days of in vitro stimulation with anti-CD40 + IL-4 (C4) or anti-CD40 + IL-4 + IL-21 (C5) was assessed by ELISA. (**B** and **C**) IgE^+^ ASC were enumerated by ImmunoSpot^®^ following five days (panel B) or seven days (panel C) of in vitro stimulation. (**D**) Representative ImmunoSpot^®^ well images for a single donor (Donor 394), tested in triplicate, following five days of in vitro stimulation, as specified. * *p* < 0.05, ** *p* < 0.01, and *** *p* < 0.001.

**Figure 5 cells-09-00433-f005:**
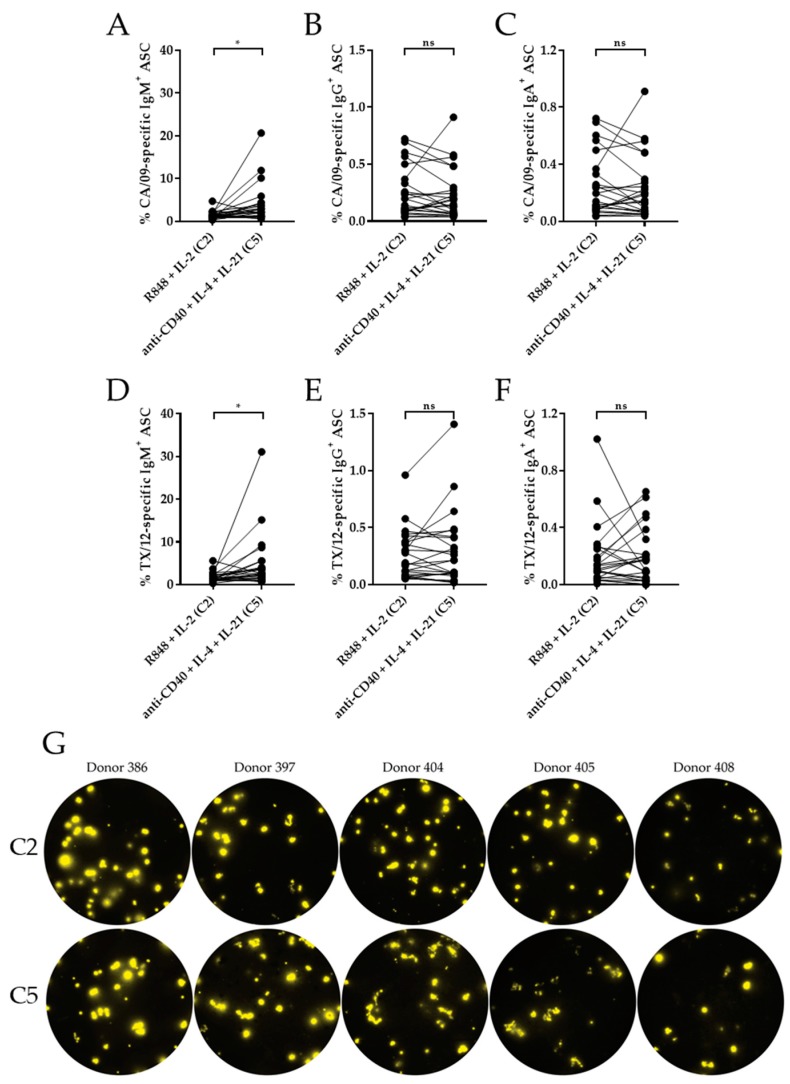
In vitro induction of influenza hemagglutinin (HA)-reactive ASC. (**A**–**F**) Healthy human donors (*n* = 24) were stimulated in vitro under two conditions, R848 + IL-2 (C2) or anti-CD40 + IL-4 + IL-21 (C5), as specified for five days and ASC were evaluated by ImmunoSpot^®^ for reactivity against recombinant hemagglutinin (rHA) of A/California/07/09 (CA/09, H1N1) (panels A–C) or A/Texas/50/2012 (TX/12, H3N2) (panels D–F) influenza vaccine strains. Frequencies of the antigen-specific IgM^+^ (A and D), IgG^+^ (B and E) and IgA^+^ (C and F) ASC following in vitro stimulation as specified above. (**G**) Representative ImmunoSpot^®^ well images of IgG^+^, TX/12 rHA-specific ASC are shown for five donors at 3 × 10^5^ PBMC per well following in vitro stimulation with R848 + IL-2 (C2) or anti-CD40 + IL-4 + IL-21 (C5) for the specified donors. * *p* < 0.05, ** *p* < 0.01, and *** *p* < 0.001.

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
