# Peer review of "IL-21 in Conjunction with Anti-CD40 and IL-4 Constitutes a Potent Polyclonal B Cell Stimulator for Monitoring Antigen-Specific Memory B Cells"

_cells, 2020, doi:10.3390/cells9020433_

Round 1
Reviewer 1 Report
The authors test several rational strategies for stimulating human B cells to quantitate polyclonal antigen-specific antibody-producing cells. The work is straight forward and important in the development of approaches that can be used to measure and follow B cell responses in humans. There are a few minor concerns/questions that might be addressed to improve clarity and better support the overall conclusions of the authors.
In the abstract, please add "the TLR 7/8 agonist" before R848. After 5 days in culture to stimulate B cells to IgM- and IgG- producing cells prior to quantitation of antibody producing cells, if one does the math from the ImmunoSpot results about 25-50% of the cells in PBMC cultures stimulated with several of the cocktails will be B cells. Is this the case? For the IgE producing cells, are results different when PBMC from atopic vs. non-atopic individuals are tested? In the experiment looking at potential memory responses, the authors have tested responses to two influenza antigens but have not included a non-specific protein antigen to which a memory response is unlikely, for instance ovalbumin or maybe an auto antigen. This would better support the contention that the responses observed are memory. Furthermore, can the time of in vitro PBMC stimulation be shortened, perhaps to 3 days, to better differentiate memory from primary responses?
Reviewer 2 Report
The manuscript „IL-21 in conjunction with anti-CD40 and IL-4 constitutes a potent polyclonal B cell stimulator for monitoring antigen-specific memory B cells“ by Franke F., et al., describes experimental in vitro approaches, which investigates the role of IL-21 in the context of the generation of antigen-specific B cells. The key message of the paper is that Th cells derived cytokines (namely IL-21) are major drivers in that process and the cocktail of a-CD40 plus IL-4 /IL-21 drives polyclonal B cell generation, which is comparable to the classical stimuli R848/IL-2.
Overall the study is well structured, the experiments are conclusive with appropriate controls, the figures are clear, the manuscirpt is well written. However, I have some concerns, and answering would in my opinion strengthen the conclusions drawn from the in vitro approaches, because this manuscirpt basically lacks acorrelate to the human situation.
Major concerns:
The authors never show IL-21 receptor on the PBMCs. They should show by qRT-PCR and flow cytometry IL-21R receptor expression. Both are important, because IL-21R mRNA levels are often very low. If they claim IL-21 is important the receptor needs to be verified. Although experiments with rHA-proteins in vivo were done, the correlate to the human situation is missing. At least one experimental setup should be added, eg. IL-21 serum levels in correlation to Ig-levels/subtypes in influenca patients, or freshly vaccinated people. Or the amount of IL-21+Th cells in relation to Ig-levels, Ig+ ASC, or others. I will not dictate an experiment, based on the availibility of human material to the group. But, the authors should show a human correlate. Does anit-CD40 /IL-4 stimulation upregulate IL-21R on PBMC. This is an important question, which may answer, why the a-CD40, IL-4, IL-21 cocktail does these stimualtions. The authors need to show this experiment to draw a conclusion. IL-21 is a major cytokine of Th17 cells. Do other Th17 cytokines show similar effects? What about IL-17A? An important conlcusion would be that IL-21 does and another cytokine does not, or that IL-21 does the same compared to IL-17A. The authors should comment on this based on a couple of available publications and ideally repeat some of the key findings with another Th17 cytokine (IL-17A, F....).
Minor concers:
Introduction: line 97 until 125. The description of the technical methods should be transfered to the materials/methods section The authors use often the term immunomonitoring. What is the exact meaning - especially in their setting? And why do we need this immunomonitoring in the setting of influenca? IL-21 has protumoral (colitis associated cancer, pancreas cancer) and antitumoral (melanoma, lymphoma) abilities. Also B cells where described to promote cancer or to inhibit cancer, depending the type of tumor and the setting. A section in the discussion should be added, regarding these facts in context to the recent literature, and how the presented results may be important in a tumor setting.Author Response
Please see the attachment.

Round 2
Reviewer 2 Report
The authors addressed a variety of points by adding new references, discussion and also changes in the manuscript. I would agree, if they skip point 2 (beyond scope) but still two major Points should be shown, because otherwise the clonclusions are still on a ground which is not too solid. The whole study is based on IL21 - why do the authors avoid to show IL21R is expressed on THEIR cells? This is a very important point.
Please address the two Points raised in my fist review by adding Experiments.
“The authors never show IL-21 receptor on the PBMCs. They should show by qRT-PCR and flow cytometry IL-21R expression. Both are important, because IL-21R mRNA levels are often very low. If they claim IL-21 is important the receptor needs to be verified.”
This is absolutely necessary. It must be shown that the cells, they worked with and built up their whole study that the receptor is expressed!
“The authors never show IL-21 receptor on the PBMCs. They should show by qRT-PCR and flow cytometry IL-21R expression. Both are important, because IL-21R mRNA levels are often very low. If they claim IL-21 is important the receptor needs to be verified.”
The authors stated that this point is pivotal. They should do an easy experiment and stimulate the PBMC to detect upregulation. Because then their conclusions will become more fundament.
Experiments from point 1 and 3 can easily be done in one setting stimulated vs. unstimulated and jsut see1.) IL21R is expressed 2.) IL21R is regulated.
